# Functions of Steroid Hormones in the Male Reproductive Tract as Revealed by Mouse Models

**DOI:** 10.3390/ijms24032748

**Published:** 2023-02-01

**Authors:** William H. Walker, Paul S. Cooke

**Affiliations:** 1Department of Obstetrics and Gynecology, Magee Womens Research Institute, University of Pittsburgh, Pittsburgh, PA 15260, USA; 2Department of Physiological Sciences, University of Florida, Gainesville, FL 32611, USA

**Keywords:** androgen receptor, estrogen receptor, glucocorticoid receptor, Sertoli cell, Leydig cell, peritubular myoid cell, testis, prostate, seminal vesicle, epididymis

## Abstract

Steroid hormones are capable of diffusing through cell membranes to bind with intracellular receptors to regulate numerous physiological processes. Three classes of steroid hormones, namely androgens, estrogens and glucocorticoids, contribute to the development of the reproductive system and the maintenance of fertility. During the past 30 years, mouse models have been produced in which the expression of genes encoding steroid hormone receptors has been enhanced, partially compromised or eliminated. These mouse models have revealed many of the physiological processes regulated by androgens, estrogens and to a more limited extent glucocorticoids in the testis and male accessory organs. In this review, advances provided by mouse models that have facilitated a better understanding of the molecular regulation of testis and reproductive tract processes by steroid hormones are discussed.

## 1. Androgen Actions in the Male Reproductive Tract

Androgen is essential for the development of testes and the male reproductive tract, as well as maintenance of male fertility. Testosterone produced by Leydig cells is the major androgen responsible for androgenic effects in the testis and the male reproductive tract that has access to testosterone in fluid derived from the testis. In male accessory organs not exposed directly to testicular fluid, conversion of plasma-derived testosterone to dihydrotestosterone (DHT), the reduced and more potent form of testosterone, contributes to androgen actions. Androgens are ligands for androgen receptor (AR, NR3C4), a 110 kDa intracellular protein encoded by a single gene on the X chromosome. Upon androgen binding, AR is released from sequestering heat shock proteins in the cytoplasm and either translocates to the nucleus to regulate gene expression or to the plasma membrane to activate kinase signaling pathways. In the absence of androgen or AR, male accessory organs do not develop. In the testis, lack of testosterone or AR causes spermatogenic arrest during meiosis, resulting in infertility. Global and cell-specific alteration of AR expression has provided important information revealing the mechanisms by which androgen and AR support organ development and function [1].

## 2. Global Elimination of ARs

### 2.1. ARKO Mice

Because one copy of the *Ar* gene is present on the X chromosome, mice with a naturally occurring dominant testicular feminization mutation (*Tfm*) in the *Ar* gene provided a male global AR knock out model that was studied for 30 years before inducible gene elimination technology was developed [2]. Finally, in the early 2000s AR knock out mice were produced. Mice with floxed AR exons mated with mice carrying a CRE recombinase transgene (*Cre*) driven by the *β-actin* or *phosphoglycerate kinase* (*PGK*) promoter resulted in global AR knockout (ARKO) mice lacking AR in all cells [3,4]. ARKO and *Tfm* male mice have an external female-like appearance with anogenital distance similar to females, indicating a lack of androgen activity to provide for masculinization during fetal development. Structures derived from the Wolffian ducts (ductus deferens, epididymis, and seminal vesicles) or urogenital sinus (prostate, bulbourethral glands) are absent. Fallopian tubes and uterus are not observed, and no vaginal opening is found in ARKO males, but *Tfm* male mice have a short blind-ending vagina, similar to what is seen in humans with natural mutations of AR. Testes are 20% of wild-type size and not descended into the scrotum. No haploid germ cells are present and there are few meiotic germ cells. These mouse models confirmed earlier androgen ablation studies showing that testosterone signaling was required for spermatogenesis and male fertility, but cell-specific mechanisms were not identified (reviewed in [5]). A summary of mouse models that provided molecular, cellular and physiological information about the development and regulation of the testis and the regions of the male reproductive tract is provided in Table 1.

### 2.2. Inducible AR Knock Out (iARKO)

A different strategy to investigate AR functions in adults was employed to develop the iARKO mouse, in which *Ar* exon 2 was eliminated using a tamoxifen-inducible Cre recombinase [6]. After 5 days of tamoxifen treatment, weights of iARKO mouse androgen target organs were reduced, including those for epididymis, seminal vesicles and anterior prostate. Organ weight reductions continued to be observed for testes, ventral prostate and kidney 11 days after the first tamoxifen treatment. Spermatogenesis and AR-dependent transcription in Sertoli cells was unaltered 5 days post-treatment, but functional markers of AR activity were impaired at later timepoints such that AR-mediated gene expression was less than 10% of normal and sperm production was essentially absent 50 days after the 5-day tamoxifen regime was completed. One drawback of the iARKO model is the off-target effects of tamoxifen, which acts as an estrogen agonist at the hypothalamic, pituitary and Leydig cell levels to alter serum FSH, LH and testosterone concentrations. Altered concentrations of these fertility-regulating hormones confounds interpretation of results from iARKO mice and emphasize the need for an alternative inducible AR knock out strategy.

## 3. Sertoli Cell-Specific Mouse Models

### 3.1. Sertoli Cell-Specific Knockout (SCARKO)

The use of Cre transgenes driven by cell-specific promoters to produce conditional knockout mice allowed identification of AR actions in specific cells and tissues. Initial studies employing selective knock out of AR in Sertoli cells using Cre recombinase expressed from the Sertoli cell specific anti-Mullerian hormone (AMH) promoter allowed normal development of urogenital tracts, and normal testis descent into the scrotum. Germ cell entry into meiosis was normal. However, spermatogenesis was arrested during the pachytene or diplotene stages of meiosis prior to the first meiotic division. Additionally, fewer seminiferous tubules displayed lumens, indicating that fluid secretion by Sertoli cells was impaired [7,8]. Testis size was reduced to 28% of normal in SCARKO mice with spermatocytes, spermatids and elongated spermatids reduced to 64%, 3% and 0% of normal, respectively. These results demonstrated the importance of Sertoli cell AR signaling for the support of spermatogenesis. Sertoli cell numbers were essentially normal in SCARKO mice [4]. In one SCARKO model, follicle-stimulating hormone (FSH) levels were elevated, but luteinizing hormone (LH) and testosterone levels were not significantly different from normal [4]. In another model, testosterone levels were decreased [8].

Follow-up studies of SCARKO mice revealed additional structures and processes in Sertoli cells needed to maintain fertility require androgen signaling. Formation of the blood–testis barrier (BTB), including correct temporal expression and localization of tight junction proteins to regions of cell–cell adhesion, was delayed and decreased in SCARKO mice. Eventually, the BTB formed in about 85% of tubules of adult SCARKO mice [29], but regions of permeability remained within the tubules. Defective barrier formation was accompanied by delayed or incomplete Sertoli cell maturation and polarization as well as aberrant positioning of the nucleus and cytoskeletal elements. Together, these findings from SCARKO mice demonstrate that AR-mediated structural components are required to create the specific germ cell environment necessary for normal development [29]. A summary of testicular intercellular communication and physiological processes required for male reproduction regulated by steroid hormone receptors that were identified from mouse models is provided in Figure 1.

The critical necessity for AR activity in Sertoli cells was confirmed using comparisons of hypogonadal (hpg) mice lacking circulating gonadotropins and having low androgen levels versus hpg.ARKO or hpg.SCARKO mice having the additional absence of AR in all cell types or specifically in Sertoli cells, respectively. These models were used to show that androgen stimulation of spermatogonia and spermatocyte development requires direct action on Sertoli cells because androgen induced seminiferous tubule and germ cell development only in hpg mice [30].

### 3.2. Hypomorph SCARKO Mice

The Braun lab produced an *Ar* null mutant in Sertoli cells that resulted from an inversion of exon 1. The mouse had a phenotype similar to SCARKO mice lacking exon 2 or 3. A hypomorph mouse line was also produced in which AR activity was reduced in Sertoli cells due to the insertion of the neomycin gene into exon 1 of the *Ar* gene. Expression of the *Rhox5* testosterone activity marker was reduced by 80% in the *Ar* null model but a 45% reduction in expression of the testosterone activity marker was observed in the hypomorph suggesting that partial AR activity was retained. In contrast to models lacking *Ar* exons, the hypomorph model was able to maintain meiosis (although at reduced efficiency) suggesting that meiosis can be completed despite reduced AR-mediated signaling. Production of spermatids in the hypomorph model allowed studies of testosterone signaling on spermatid development. These results confirmed earlier hormone deprivation studies showing that AR function in Sertoli cells is required for post-meiotic functions including adhesion of spermatids to Sertoli cells during their initial stages of elongation and release of mature elongated spermatids from seminiferous epithelium [9].

### 3.3. Gain of Function, TgSCAR

Effects of precocious AR expression in Sertoli cells were investigated in the transgenic Sertoli cell androgen receptor (TgSCAR) model mouse in which an AR transgene was expressed specifically in Sertoli cells using the androgen-binding protein (*Abpa*) gene promoter. In these mice, AR expression was much stronger than normal in Sertoli cells two to five days after birth when AR production is usually first detected. TgSCAR mice had testes up to 40% smaller than normal due to a reduction in the numbers of post-natal Sertoli cells at each developmental stage from post-natal day (PND) 5 to 70. Tubule lumen formation (a measure of Sertoli cell maturation) occurred earlier and expression of AR-mediated gene expression was greater in TgSCAR testes. The ratio of spermatocytes and spermatids to Sertoli cells was also higher at earlier developmental timepoints (PND 10-30) indicating accelerated spermatogenic development. These results support the idea that precocious AR expression causes earlier Sertoli cell maturation in the TgSCAR mouse [10].

### 3.4. Specificity-Affecting AR Knock-In (SPARKI)

Information about AR interactions with androgen response elements (AREs), DNA sequences regulating gene expression, was provided by the SPARKI mouse model [11]. This mouse has AR exon 3 encoding the second zinc finger replaced with exon 4 of the glucocorticoid receptor (GR). The chimeric protein retains the ability to bind classic androgen response elements (AREs) that are bound and regulated by AR, GR and progesterone receptor but, not selective AREs that are bound and regulated only by AR. AR expression levels in androgen target tissues was normal, but SPARKI mice had decreased fertility with litters reduced 2- and 4-fold in size and frequency. Testis, epididymis, ventral prostate, anterior prostate and seminal vesicles had reduced weights (55–67% of normal). Testes in SPARKI mice were descended into the scrotum, but Sertoli and germ cell numbers were reduced to 68% and 55% of controls, respectively. Reduction of Sertoli cell numbers was also observed in ARKO but not SCARKO mice, suggesting an androgen-dependent process outside of the Sertoli cells that requires selective AREs contributes to determining Sertoli cell numbers. SPARKI mice were able to support spermatogonia normally, but fewer spermatocytes and spermatids were present per Sertoli cell, indicating a decreased Sertoli cell ability to support maturation of meiotic and post-meiotic germ cells. Expression of some AR-dependent mRNAs found in SCARKO mice (*Eppin*, *PCI*, *Tubb3*) was not altered whereas, other mRNAs (*Rhox5*, *Tsx*, *Drd4*) were severely reduced. These results indicate that the SPARKI model can be used to differentiate members of the two classes of AR regulated genes.

### 3.5. Glucocorticoid Receptor Knock Out

The SPARKI mouse model results suggested that glucocorticoids acting via GR can contribute to regulation of some AR-mediated gene expression in Sertoli cells. Consistent with this idea, the Sertoli cell GR knock out (SCGRKO) mouse model provided evidence that signals downstream of GR may modulate functions supporting spermatogenesis. Weights of testes, epididymis and seminal vesicles were normal and SCGRKO males were fertile with normal litter sizes. However, the percentage of patent seminiferous tubules was reduced, and tubular diameters were inexplicably increased in SCGRKO mice. Numbers of Sertoli cells, spermatocytes and round spermatids were similarly reduced (25–30%) with Sertoli-germ cell ratios remaining normal. Leydig cell numbers were normal but there was Leydig cell hypertrophy, accumulation of lipid droplets in some Leydig cells, altered expression of some Leydig cell transcripts required for steroidogenesis and a reduction of *Lhr* mRNA, which might account for the 25% reduction of testicular androgens. These results suggest that signals downstream of GR in Sertoli cells partially enhance Leydig cell function and reinforce the idea that Sertoli cells communicate with and maximize effectiveness of Leydig cells. Because GR can be detected in post-natal but not adult Sertoli cells, the SCGRKO phenotype may represent developmental defects that lead to lasting effects [12].

## 4. AR-Regulated, Differentially Expressed Genes (DEGs) Identified by Mouse Models

### 4.1. SCARKO Models

Early studies of AR-regulated genes identified using SCARKO mice were reviewed previously [5,31]. These studies characterizing gene expression in whole testis extracts revealed large sets of differentially expressed mRNAs but few that were altered more than 2-fold across all mouse models. Categories of regulated genes included proteases, protease inhibitors, cell adhesion and cytoskeletal proteins, supporting the idea that testosterone signaling targets cell–cell junctional dynamics and restructuring of the seminiferous tubules required for germ cell adhesion and migration. mRNAs encoding solute carriers and transport proteins (transferrin, fatty acid binding protein and ABP) as well as modifiers of metabolism were differentially expressed in some AR-deficient models, suggesting that testosterone contributes to creating a specific unique environment needed for germ cell development. There was also differential expression of genes involved in vitamin A metabolism/retinoic acid production that are required for differentiation of spermatogonia. In addition, Sertoli cell AR was found to regulate mRNAs encoding proteins required for structural changes to the surfaces of germ cells and spermatozoa that are needed to maintain fertility.

A follow-up study of the AR hypomorph models using mouse genome arrays detected 62 transcripts in *Ar* mutant testes that differed by more than 2-fold, with about twice as many being downregulated by AR as upregulated. Genes involved in metabolic processes and signal transduction were identified as major responders to the loss of AR activity. About 65% of the AR regulated genes were associated with an ARE within 6 kb of their transcription start sites and 23% were previously shown to be regulated by AREs suggesting that many AR-regulated genes are directly regulated by AR binding to AREs [32].

### 4.2. Ribotag Mouse Models

Ribotag mice harbor an *Rpl22* transgene encoding a ribosomal protein that is tagged with a hemagglutinin (HA) epitope in response to expression of Cre recombinase. This strategy allowed isolation of ribosome-associated (actively translated) mRNAs in specific cell types in the presence or absence of AR expression. Using Leydig cell-specific (Cyp17iCre) or Sertoli cell-specific (AMH-Cre) transgenes with intact endogenous AR expression and a microarray approach for mRNA analysis, numerous LH-regulated, ribosome-associated mRNAs were identified in Leydig cells. However, few differentially regulated translated mRNAs were found to be enriched in Sertoli cells of adult mice after a four-hour stimulation with testosterone [13].

Using RiboTag mice having AR knocked out in Sertoli cells and an RNA-seq assay allowed the identification of a large repertoire of new and previously undetected AR-regulated translated mRNAs in PND 10 Sertoli cells and indirectly regulated mRNAs in other cells. Approximately 1000 mRNAs were found to be differentially expressed after elimination of Sertoli AR, with slightly more down-regulated than up-regulated in total testis. However, Sertoli cell-specific mRNAs were more often down-regulated by testosterone via AR because 60% of Sertoli mRNAs regulated 2-fold or more are up-regulated in the absence of Sertoli AR [14]. This finding supports most other gene expression studies that found testosterone and AR down-regulate more mRNAs than they upregulate [5]. Differentially expressed mRNAs were enriched for the following functions: calcium ion binding, cytoskeleton protein binding, biological adhesion, cell–cell junction and anchoring, stereocilium, oxidative stress protection, metabolism/detoxification and protease/antiprotease balance [14].

### 4.3. Single Cell RNA Sequencing (scRNA-Seq) Analysis of SCARKO Mice

Employment of scRNA-seq technology with SCARKO mice added to information provided by the RiboTag mouse models about Sertoli cell- and germ cell-specific genes and pathways that are altered downstream of testosterone and AR signaling. Using scRNA-Seq and 20-day-old SCARKO mice, 347 up-regulated and 250 down-regulated genes expressed in Sertoli cells were identified, again indicating that most DEGs were downregulated by testosterone and AR signaling. However, it was not yet known whether AR directly down-regulates a larger population of genes or whether many testicular genes are down-regulated downstream of initial AR-mediated gene expression changes in Sertoli cells. At least 100 of the identified DEGs are known be associated with spermatogenesis, apoptosis, meiotic cell cycle, reproductive processes, male gamete generation germ cell development and cell migration. Transcriptome analysis showed that expression of cell migration/growth genes is a major alteration occurring in the absence of AR that may be an important function required for productive Sertoli cell signaling. Pseudotime trajectory analysis indicated that Sertoli cell development and maturation was affected by the loss of AR. The scRNA-seq strategy determined that lack of AR in Sertoli cells did not decrease the number of differentiated spermatogonia developing from germ line stem cells but, expression of over 100 spermatogonia mRNAs were altered including those associated with germ cell development and spermatogenesis [33].

## 5. Regulation of Germ Cell Meiosis by AR Activities in Sertoli Cells

The Handel group used scRNA-seq and SCARKO mice in which all licensed germ cells were synchronized to enter meiosis concurrently to examine the regulation of meiotic progression by Sertoli cell AR [34]. These studies of SCARKO mice revealed that necessary double strand DNA breaks were produced normally early in meiosis, and repair of the breaks was similar to wild type. Synaptonemal complex formation that allows pairing of homologous chromosomes and the production of chiasma required for recombination also occurred in SCARKO mice. Loss of AR in Sertoli cells did not impede initiation of chromosome synapsis or recombination events, but later condensation of chromosomes and entry into metaphase was absent.

Combining a scRNA-seq strategy with germ cell synchronization in SCARKO testes, it was found that differential gene regulation in germ cells begins in differentiated spermatogonia and leptotene/zygotene spermatocytes. Transcriptomic arrest occurs in most SCARKO germ cells during leptotene/zygotene. However, cell development continues to a mid-pachytene state, implying that cytological progress is uncoupled from the molecular gene regulatory environment. These results indicate that androgen signaling from Sertoli cells is critical throughout early meiotic prophase and for transition to the divisional phase of meiosis. Down-regulated genes identified in SCARKO testes included those associated with cytoskeleton, acrosome or cilia organization and formation that are required for post-meiotic germ cell development. This finding raises the question of whether the lack of post-meiotic spermiogenesis gene expression is a contributing signal to meiotic arrest or simply results from germ cells that are not destined for further development. Upregulated SCARKO mRNAs included those for mitochondrial membrane and oxidative phosphorylation, which may be due to germ cell disruption or decreased ability to repress oxidative phosphorylation in SCARKO testes. A relatively small number (165) of Sertoli cell mRNAs were differentially regulated by more than 2-fold. Of the 165 DEGs, 77 were previously reported in the RiboTag/SCARKO mice study [14] that did not have the advantage of scRNA-seq technology [34]. The small number of differentially DEGs was in agreement with the lack of global transcriptome differences observed between the wild-type and SCARKO Sertoli cell populations in this study [34] but, the transcriptome changes did not agree with dramatic differences in the transcriptomes of wild type and SCARKO Sertoli cells observed by Cao and colleagues [33].

The Liu lab used SCARKO mice to study processes in meiotic germ cells that require androgen-mediated signals from Sertoli cells [35]. In SCARKO mice, chromosomal synapsis during meiosis was aberrant with unsynapsed and partially synapsed chromosomes as well as univalent chromosomes, indicating that a signal from Sertoli AR is required to complete chromosomal synapsis. As shown by the Handel group, double strand breaks in chromosomes, a process that occurs early in meiosis and is required for recombination, was found to occur normally by Liu and colleagues in SCARKO mice. However, meiotic recombination and repair of chromosomal breaks was disrupted at least in part by abolishing recruitment of *Rad51* recombinase to DNA breaks. Absence of recombination and repair and the incomplete synapsis of chromosomes are known to activate cell checkpoint responses in meiotic germ cells that act later to halt meiosis during the pachytene stage prior to the first meiotic division [36,37,38,39].

SCARKO testes were also found to express elevated levels of EGF family growth factors, suggesting that signaling downstream of AR normally limits production of EGF family members. The finding that an inhibitor of EGF receptor activity partially rescued meiotic progression in EGF-overexpressing mice provided more support for the idea that androgen-mediated limitation of EGF production during specific stages of germ cell development may be required to maintain meiosis [35]

## 6. Androgen Pathway Selective Transgenic Mice

Steroid hormones act by at least two intracellular mechanisms. In the classical or genomic pathway, steroid hormones bind their specific receptors, which dimerize to form a DNA binding complex that regulates gene expression. In the nonclassical or nongenomic pathway, steroid hormones act via receptors associated with the internal side of the plasma membrane to modulate kinase pathways to directly alter cellular processes, such as cAMP concentrations, Ca^2+^ concentrations and the PI3K/AKT and MAPK/ERK protein kinases, as well as gene expression [5].

Understanding relative functions of classical vs. nonclassical testosterone signaling in Sertoli cells was impeded by a lack of experimental systems to identify testosterone actions mediated by each pathway. To study contributions of each signaling pathway in vivo, unique transgenic mouse models were produced that expressed a wild-type AR transgene or transgenes having independent mutations incorporated into the AR transgene [15]. In the AR-Classical (AR-C) transgenic model, classical nuclear androgen signaling was retained, but nonclassical AR signaling was absent. In AR-Nonclassical (AR-NC) transgenic mice, nonclassical AR signaling was retained, but classical signaling was absent. The transgenes, controlled by regulatory regions that flank the genomic *Ar* gene, were expressed in a correct temporal and cell-specific manner and did not disrupt spermatogenesis or fertility in the presence of endogenous AR [15,40].

To study functions of individual testosterone pathways in Sertoli cells, endogenous AR was knocked out specifically in these somatic cells on gestation day 16.5 (GD 16.5) using AMHCre. Testis weights of mice lacking AR in Sertoli cells but having both AR-C and AR-NC or AR-C alone were 87% and 81% of wild type, respectively. However, AR-NC expression alone did not alter testis weight (26% of wild-type) above that of SCARKO mice. The gross appearance of other reproductive tract tissues including seminal vesicles (an indicator of functional testosterone signaling) were unaffected by the Sertoli cell-specific changes in AR activity. Spermatogenesis was not disrupted in mice expressing both AR-C and AR-NC. In AR-C mice, spermatogenesis could be completed. However, 50% of AR-C seminiferous tubules showed evidence of inefficient spermatogenesis, including tubules with no or prematurely released germ cells, tubules lacking lumens, disorganized germ cell positioning and mis-localization of elongated spermatids. These findings suggest that AR-C requires additional AR-NC activity for fully efficient spermatogenesis and support of germ cell development, differentiation and motility including the proper migration of elongated spermatids. Spermatogenesis could not be completed but was maintained up to the round spermatid developmental stage in 36% of AR-NC mice, indicating that AR-NC activity in Sertoli cells contributes to the completion of meiosis and that AR-C is required for elongated spermatid development. Together, the pathway-specific AR transgene studies show that AR-C and AR-NC testosterone pathways synergize in Sertoli cells to permit fully efficient spermatogenesis and fertility [15].

A survey of germ cell mRNAs essential for completion of meiosis in SCARKO and SCARKO + AR-C testes revealed that AR-C was required for expression of three mRNAs (*Syce1*, *Syce2* and *Syce3*) encoding essential components of the synaptonemal complex central element and 2 members of the cohesion complex (*Rec8*, *Stag3*) needed for synaptonemal complex formation and completion of meiosis. In agreement with Larose and colleagues [34] the synaptonemal complex was observed to form in SCARKO mice but, new studies showed SYCE protein expression was reduced along synapsed homologous chromosome pairs [15]. Decreased expression of synaptonemal complex components is known to disrupt chromosome synapsis, inhibit later recombination and DNA repair, and elicit later cell checkpoint signaling responses resulting in meiotic arrest and apoptosis of pachytene spermatocytes [36,37,38,39]. Together, these findings provide a long-sought mechanism for the delayed meiotic arrest that occurs in the absence of testosterone signaling and SCARKO models.

Overall, studies of AR-C and AR-NC transgenes on a SCARKO background determined that: (1) the two signaling pathways had complementary functions that together provided optimal male fertility, (2) nonclassical signaling contributes to BTB maintenance, spermatid adhesion plus migration and localization of elongated spermatids and (3) classical signaling contributes to BTB maintenance, transition of round to elongated spermatids and elongated spermatid development and is required to complete meiosis including inducing mRNAs required for chromosome synapsis [15].

In mice with global knockout of endogenous AR, restoration of AR-C signaling alone provided 76% of normal androgen activity during fetal development as measured by relative anogenital distance, whereas AR-C plus AR-NC together provided for full androgen-dependent development. Testosterone signaling through AR-C plus AR-NC or AR-C alone, but not AR-NC alone, were sufficient for production of mature elongated spermatids/sperm. However, none of the mice lacking endogenous AR were fertile because efferent ducts, epididymis and ductus deferens did not develop and thus sperm delivery was not possible. This result suggests that development of these accessory reproductive organs is more sensitive to testosterone signaling than is the testis. Testosterone-dependent prostate lobe development was partially supported by AR-C, but AR-C plus AR-NC synergism supported 2-fold additional development of prostate lobes to 57% to 74% of normal [15]. These data provided additional evidence of cooperativity between AR-C and AR-NC signaling pathways.

## 7. Peritubular Myoid (PTM), Leydig and Germ Cell-Specific AR Knockout Models

### 7.1. PTM Cells

Sertoli cells are not the only critical somatic cell type supporting spermatogenesis, as shown by results from the PTM-ARKO mouse model that eliminated AR from smooth muscle cells including PTM cells surrounding the seminiferous tubules and in blood vessels. Although Cre recombinase expression driven by the smooth muscle myosin heavy chain (Myh6) promoter could be detected in only about 40% of PTM cells, male PTM-ARKO mice did not sire any pups, became azoospermic and progressive loss of spermatogonia and spermatocytes was more dramatic than observed in adult SCARKO mice. Sertoli cell numbers did not vary between PTM-ARKO and SCARKO mice but expression of AR-dependent mRNAs in Sertoli cells was decreased and Sertoli cell nuclei orientation in PTM-ARKO testes became progressively more disorganized, with Sertoli cells often mis-localized in the middle of the seminiferous tubules by PND 100. One possible mechanism responsible for disruption of Sertoli cell polarization and attachment to the basement membrane in PTM-ARKO mice may be due to disrupted deposition of extracellular matrix by PTM cells, that in turn alters Sertoli cell function [16].

A second mouse model, PM-AR^−/y^, with AR excised in PTM cells used the Cre transgene driven by the transgelin (Tagln) smooth muscle protein 22-a promoter. This model had normal fertility although testes weight was reduced 34%, total germ cell numbers decreased and epididymal sperm numbers were reduced 57%. Expression of Sertoli cell marker genes including those required for cell–cell interactions and genes required for PTM cell contraction was decreased. Reduced sperm production in PM-AR^−/y^ and PTM-ARKO mice may partially reflect altered Sertoli cell gene expression [17]. The less dramatic PM-AR^−/y^ fertility phenotype may be due to lower efficiency of the smooth muscle protein 22-a driven Cre transgene in the testis [16]. The strong phenotypes of the PTM-ARKO models suggest that PTM cells provide important contributions to maintain fully efficient spermatogenesis and highlight the importance of PTM-Sertoli cell communication.

In PTM-ARKO mice, Leydig cell numbers and AR expression were not altered but intratesticular (but not serum) testosterone levels were increased. LH levels were increased, perhaps as a compensatory mechanism to stimulate testosterone production from Leydig cells [16]. Further examination revealed that PTM-ARKO Leydig cells undergo initial differentiation from progenitor to immature Leydig cells, but do not complete their maturation. In adults, there were two roughly equal sized populations of Leydig cells. Both populations had larger, irregularly shaped mitochondrial evidently lacking the outer mitochondrial membrane. However, one population expressed normal levels of Leydig cell markers and steroidogenic enzymes, whereas a second “abnormal” population had arrested development with weak expression of insulin like 3 (INSL3), LH receptor, several steroidogenic enzymes and markers of Leydig cell differentiation as well as abnormal accumulation of lipid droplets. These results suggest that AR in PTM cells is required to support Leydig cell maturation and that after elimination of AR in PTM cells, the “normal” population of Leydig cells must produce more testosterone to compensate for overall decreased expression of steroidogenic enzymes and lower testosterone production from abnormal Leydig cells. The PTM-ARKO model also provides evidence that like Sertoli cells, there is paracrine signaling from PTM cells required for Leydig cell differentiation and function [41].

### 7.2. Leydig Cells

Leydig cell AR knock out (L-AR^−/Y^) mice showed that functional AR in Leydig cells is essential to maintain spermatogenesis, testosterone production and fertility. Using an AMH receptor-Cre (AMHR-Cre) strategy to eliminate AR in Leydig cells and partially in Sertoli cells, two patterns of seminiferous tubules were observed. One showed a maturation arrest phenotype with significant deficiency of AR in both Leydig and Sertoli cells. A second Sertoli cell-only phenotype lacking germ cells was also observed in which most Leydig cells were negative and Sertoli cells showed relatively weak AR immunoreactivity [8,18]. The presence of AR-positive Leydig cell clusters indicated that Cre recombinase activity was not fully efficient [42]. Together, the less specific and efficient knock out of AR in the L-AR^−/Y^ model and the loss of AR in some Sertoli cells limits interpretation of the results.

L-AR^−/Y^ mice were azoospermic with fewer proliferating spermatogonia and decreased numbers of apoptotic pachytene spermatocytes but with the majority of testes showing germ cell development halted during the second spermatocyte or early round spermatid stages of development. Expression of mRNAs encoding key steroidogenic enzymes was reduced, including 17β-HSD3, 3β-HSD6 and P450c17, resulting in hypotestosteronemia that contributes to blocking completion of spermatogenesis [42].

In ARKO and SCARKO mice, fetal Leydig function is normal. However, there were 60–83% fewer Leydig cells in adult ARKO mice due to attenuated increases in initial expansion of the adult Leydig cell population between PND 5 to 20. The size (surface area) of ARKO Leydig cells did not increase [43,44]. Although one study reported no difference in Leydig cell numbers in SCARKO mice [45], others found adult Leydig cells were reduced by 40% in SCARKO mice, and the cells were larger with more lipid droplets and mitochondria than control mice [43,44]. Steroid production appears to be increased to compensate for Leydig cell reductions as Leydig cell mRNAs required for the steroidogenesis pathway were increased in SCARKO testes. Expression patterns of steroidogenesis mRNAs in ARKO mice are more complex, with some mRNAs encoding P450scc and 3β-HSD-I increasing, and some significantly reduced or barely detectable such as 17β-HSD III, 3β-HSD VI, PGD and P450c17 [43,44]. Decreased Leydig cell numbers in SCARKO and ARKO mice may be due to lower PDGF-A levels, which was proposed as a Leydig cell growth factor during development [7]. Together, the mouse models provide evidence that androgen signaling including AR actions in Sertoli cells are required for expansion and maturation of adult Leydig cells.

A scRNA-Seq study of SCARKO mice also determined that numbers of mature Leydig cells were reduced, whereas interstitial progenitor and immature Leydig cells accumulated [34]. This study indicating that the population of all Leydig cells remains similar to normal is in agreement with earlier results showing unchanged Leydig cell levels using antisera against Cyp11A1, which detects both immature and mature Leydig cells [44]. Further evidence of Sertoli cell influence on Leydig cell maturation and expansion was shown in the TgSCAR model mouse in which an AR transgene is precociously expressed specifically in Sertoli cells. TgSCAR mice had lower proliferation and reduced numbers of postnatal fetal and adult Leydig cells. mRNAs encoding members of three Sertoli-Leydig paracrine signaling pathways (*Dhh*-*Ptch1*, *Pdgfa*-*Pdgfra*, *Amh*-*AMhr2*) are induced in TgSCAR mouse testes. Intratesticular testosterone levels per Leydig cell were elevated in TgSCAR mice [12]. Results from the TgSCAR model demonstrated that differentiation, steroidogenic function and size of the Leydig cell population are controlled downstream of Sertoli cell AR.

Comparisons of hpg, hpg.ARKO and hpg.SCARKO mouse models showed that signals from testicular ARs during fetal development inhibit an inappropriate LH-dependent developmental pathway in Leydig/adrenal cell precursors that results in cells with adrenal characteristics. This result may provide one explanation for Sertoli cell AR-mediated support of the Leydig cell population. The hpg-based models also showed that LH acts to induce the steroidogenic potential of Leydig cell precursors but additional stimulation via ARs is essential for induction of the Leydig cell phenotype [46].

### 7.3. Germ Cells

Excision of AR during early meiosis in leptotene-zygotene germ cells using the Sycp1-Cre transgene in the G-AR^−/y^ mouse model did not alter sperm count, fertility, seminiferous tubule structure or the relative numbers of germ cells at various stages of development. This mouse model provides the important finding that meiotic and post-meiotic germ cell AR, if expressed, is not required for spermatogenesis or male fertility [8].

## 8. Knockout Mouse Models Reveal Collaborative Interrelationships between Sertoli, PTM, Leydig and Germ Cells

Together, results from the knockout mouse models revealed that there is an interdependence between testicular cells that is required for cell maturation and maintenance of spermatogenesis. Testosterone-mediated signals from Sertoli and PTM cells contribute to proliferation and maturation of adult Leydig cells while facilitating steroidogenesis. PTM cells provide extracellular matrix and other cell–cell contact signals to improve Sertoli gene expression and polarization. Finally, AR-mediated signals from Sertoli cells support maintenance of the blood–testis barrier, meiosis, spermatid attachment and sperm release (Figure 1).

## 9. Knockout of AR in Accessory Sex Organs

### 9.1. Prostate

Functions of AR in epithelial cells of prostate were examined using two similar mouse models. These models employed a composite probasin promoter (*Arr2pb*) including two androgen response elements (AREs) from the promoter to drive Cre expression and selectively eliminate AR expression in prostate epithelium as well as epididymis, seminal vesicles and ductus deferens. Loss of AR in prostate epithelium was progressive, becoming nearly total by 24 weeks postnatal. In the prostate epithelium ARKO (PEARKO) mouse model, the macroscopic prostate structure appeared normal but prostate lobe weights were uniformly reduced [19]. Epithelial branching remained intact in PEARKO mouse prostate lobes, indicating that signaling downstream of AR in prostate stroma is sufficient to maintain development of prostate and branching of epithelium [19]. In the prostate epithelial-specific ARKO (Pes-ARKO) model, weights of the ventral but not dorsolateral or anterior prostate lobes were decreased, but there were no differences in litter sizes [20]. Progressive changes in morphology included dedifferentiation of epithelium with columnar and reduced cell height, loss of infolding and detached cells [20] as well as cell clustering and absent epithelial cells causing a discontinuous epithelium [19]. Epithelial proliferation was increased but apoptosis remained low in Pes-ARKO mice. Necrotic epithelial cells were observed in lumens due to cell sloughing in the absence of AR signaling [20]. Together, these findings demonstrate that epithelial AR is required to maintain prostatic differentiation and homeostasis.

The effect of eliminating AR in prostate smooth muscle cells was assessed with the PTM-ARKO model. Ventral prostate weights were reduced after PND 35 and reached decreases of 50–60% at PND 100 and PND 200. Ventral prostates displayed elongated epithelial folding, epithelial cell hypertrophy, intracellular edema and diffuse epithelial cell hyperplasia leading to overcrowding and desquamation. However, the lack of increased proliferation in epithelial cells suggests that the apparent hyperplasia in PTM-ARKO mice is due to maintaining similar cell numbers in a smaller space. Observed differences in epithelial cell gene expression suggest a loss of cell identity. For example, decreased expression of the cell adhesion protein E-cadherin may disrupt epithelial differentiation and homeostasis. Abnormal infiltration of immune cells was observed in the stroma and acini of PTM-ARKO prostate, which may account for the fibrosis observed at PND 100 and stimulated cell proliferation. Results from the PTM-ARKO mouse demonstrate that AR signaling derived from smooth muscle mediates normal stromal-epithelial interactions, cell identity and function and limits hormone-dependent epithelial cell proliferation in prostate [47].

SM-ARKO smooth muscle AR knock out mice had prostates with normal gross appearance and branching morphogenesis, but with fewer epithelial infoldings into the lumens of acini and a thinner epithelium. Epithelial cell proliferation was decreased in the prostate lobes [21]. An explanation remains to be determined for opposing results in regard to proliferation, infolding and cell differentiation in SM-ARKO versus PTM-ARKO mice.

Knock-out of AR specifically in prostate stromal fibroblasts was accomplished using a fibroblast-specific protein 1 (FSP1)-Cre. In these mice, ventral prostates were smaller, epithelial cells were more cuboidal and flattened, epithelial proliferation was decreased and apoptosis increased in older mice. There was also dramatically decreased collagen deposition on the FSP1-ARKO prostate basement membrane, which may contribute to defects of epithelial development and growth. FSP1-ARKO mice also had down-regulated expression of FGF-10, FGF-7 and IGF-1 that act as prostate stromal growth factors in the prostate [22]. Together, results from FSP1-ARKO mice identify the impacts that signals from stromal fibroblasts have on epithelial cells.

A double stromal AR knock out (dARKO) mouse model was created by mating SM-Cre with FSP1-Cre mice. Offspring lacking AR in prostate smooth muscle and fibroblasts had smaller anterior prostate lobes, but ventral and dorsal lateral lobes were similar to controls. Branching structure was reduced in anterior and dorsolateral but not ventral lobes. Continuing loss of infolding was initiated by PND 56. Together, these results suggest that stromal/mesenchymal AR is essential for prenatal organogenesis and maintaining homeostatic balance of prostate in postnatal and adult stages. Proliferation of epithelial cells in 4- and 8-week-old dARKO prostates was decreased, but in 11- and 24-week-old mice the very low proliferation rates were similar to controls. Apoptosis rates in prostates through 16 weeks after birth were low and normal but, were increased above normal in 24-week-old dARKO mice. dARKO mice also had decreased height of the luminal epithelium with flattened and cuboidal epithelial cells. Thus, epithelial cell proliferation and proliferation is dependent upon stromal AR-derived signals in young mice and this signaling is required to maintain epithelial survival in adults. Cell lines derived from dARKO prostate were used to identify IGF-1, PGF and SPP1 as AR-regulated stromal paracrine growth factors [23].

### 9.2. Seminal Vesicles

In PEARKO mice, AR activity was also reduced in epithelium of epididymis and vas deferens and unexpectedly in smooth muscle cells of seminal vesicles due to induced expression of Cre recombinase by the probasin promoter in these organs. Seminal vesicle weights were reduced by 55% from decreased secretion and reduced weight of the emptied gland. Epithelial cells in the proximal region were less folded, lower and cuboidal, and contained little cytoplasm. The smooth muscle layer was reduced about 50% and disorganized. These results indicate that testosterone-mediated signals from smooth muscle cells are important to maintain normal epithelial and stromal cell function including secretion. Androgen signaling in smooth muscle may limit epithelial cell proliferation in seminal vesicles, because the number of epithelial cells undergoing mitosis in mice lacking AR in seminal vesicle smooth muscle cells was increased 3-fold, resulting in hyperplasia.

In other studies employing PEARKO mice, the numbers of copulatory plugs, which are dependent upon secreted proteins from seminal vesicles, were similar to normal 4 h after coitus but were smaller with a soft and fibrous consistency that likely is responsible for the 70% decrease in plug-positive females when assessed the morning after coitus. Fertility of PEARKO males was reduced, with litter size and number reduced. The success of in vivo fertilization was reduced almost 3-fold and in vitro fertilization was reduced at low sperm concentrations, perhaps due to a mild defect in sperm binding to the zona pellucida. Sperm numbers were unchanged, but sperm released from PEARKO cauda epididymis had increased percentages of abnormal bent tails with hairpin structures and displayed hyperactive, erratic, and whiplash-like movement of capitated sperm [48]. Disruption of seminal vesicle function after elimination of AR activity in smooth muscle provides a partial explanation for decreased fertility. However, as discussed, Sertoli cell function and germ cell development are also disturbed by knockout of AR in testicular PTMs in PEARKO mice [41].

The PTM-ARKO model also provided information about AR functions in seminal vesicles. The reproductive tract formed normally. However, in the absence of smooth muscle AR, seminal vesicle weight was reduced after PND 21. The epithelium was cuboidal with an increase in branching, and there was reduced smooth muscle thickness. These results indicate that testosterone-derived signals from smooth muscle cells are important to maintain normal seminal vesicle epithelial and stromal cell function including secretion. As in PEARKO mice, epithelial cells undergoing mitosis in PTM-ARKO were increased 3-fold, resulting in hyperplasia. Further studies revealed that 17β-estradiol (E2) caused more pronounced hyperplasia of the epithelium and increased smooth muscle thickness in the PTM-ARKO. The results indicate that in the absence of testosterone, estrogens can cause uncontrolled proliferation and hyperplasia in seminal vesicles [49].

### 9.3. Epididymis

In the PEARKO model, epididymal morphology was normal, but epididymal weight was reduced about 20% and stroma thickening was present in the corpus and cauda regions. Due to reduced androgen action in the epididymis, sperm may transit more rapidly through the caput and corpus regions, leading to sperm maturation defects and increased rates of spontaneous acrosome reaction in PEARKO sperm [19,48].

Two mouse models were used to study AR function in the proximal region the epididymis. Knock out of floxed AR in principal cells of the proximal (caput) epididymis using Cre recombinase driven by the *Rnase10* gene promoter (ProxE-ARKO) resulted in regression of the epididymal initial segment, epithelial hypoplasia and hypotrophy and epididymal obstruction beginning at PND 20-25. Blockage of fluid flow through the epididymis caused back pressure into the efferent ducts and testis resulting in atrophy of the seminiferous epithelium, orchitis, fibrosis of testicular parenchyma and azoospermia [24].

A second mouse model, the caput epithelium androgen receptor knock-out (CEARKO), employed a FoxG1 Cre to knock out AR in principal cells of the proximal epididymal epithelium [25]. In CEARKO mice, AR loss is first detected by PND 11, and the initial segment does not develop. In older mice, (PND 100) the smooth muscle layer of the epididymis is disorganized and disrupted. CEARKO mice develop the same obstructive azoospermia infertility phenotype as the mice having AR knocked out with the *Rnase10* Cre. Together, the two epididymis AR knock out models confirm the requirement for AR and androgen signaling in proximal epithelial cells to allow development and maintenance of the initial segment of epididymis as well as fluid and sperm flow from the testis through the epididymis to maintain fertility.

## 10. Estrogen Actions in the Male

### 10.1. Discovery of Estrogens and Their Presence in the Male

Androgens regulate the differentiation, development and adult function of the male reproductive tract, as described in detail in the initial part of this article. These findings have resulted in all facets of androgen production and action being a major focus of research in the area of male reproductive biology. Despite the primacy of androgen in terms of sex steroid regulation of male reproduction, a literature dating back almost a century has also clearly shown that estrogens are critical for male reproduction.

Evidence for the existence of an ovarian hormone that we now know was an estrogen was obtained in the early 1920s, and by 1940 E2 and other estrogens had been identified [24,50]. Initial studies had focused on the ovary in the female as the estrogen source, but only 4 years after the identification of the first estrogen produced by the ovary, Zondek [51] reported the existence of estrogen in stallion urine, indicating that estrogens were not confined to only females.

### 10.2. Estrogen Administration to Males Produces Deleterious Male Reproductive Effects

Only a year after the initial report of estrogens in the male, Burrows [52] reported the first effects of exogenously administered estrogen on the reproductive tract of the male. A significant amount of the literature linking estrogen and the male reproductive tract in the subsequent decades consisted of studies of this type showing deleterious effects of exogenous estrogen on adult male reproductive structure and function [53]. One notable exception to the typically deleterious effects of estrogens on males was the demonstration by Huggins and co-workers that exogenous estrogen could have beneficial effects on men suffering from advanced prostate cancer. This beneficial effect was due to the ability of the exogenous estrogen to feed back on the pituitary and hypothalamus and reduce endogenous androgen production that normally stimulated growth of the prostate cancer [54].

The discovery that developing males were even more sensitive than adults to estrogens led to the development of an extensive literature showing long-term deleterious effects of fetal or neonatal estrogen exposure on the testis and many other male reproductive organs [55]. The literature that documented exogenous estrogen effects in males clearly established that estrogens could have effects, sometimes profound, on developing and adult male reproductive organs. However, from these types of studies, it was impossible to ascertain whether estrogen played a role in the normal development and function of the male tract or if these substances exerted harmful effects when given exogenously but did so without having a normal endogenous role in males.

### 10.3. Estrogens Are Present in Males and Estrogen Receptors Are Widely Distributed in Males

A separate literature on circulating estrogen concentrations in males and the expression of estrogen receptors in the reproductive tract and elsewhere in the male that arose in parallel with the studies of exogenous estrogen effects in males provided additional data supporting an estrogen role in normal male reproduction. However, this remained speculative for years, as a definitive experimental system that allowed this hypothesis to be tested was unavailable. Estrogen is readily measurable in the peripheral blood of males from many animal species (reviewed in [53], although these levels are typically far less than circulating androgen concentrations. Similarly, estrogen concentrations in men are typically significantly lower than those seen in women of reproductive age, although estrogen concentrations in men can in some cases be comparable to those in post-menopausal women. In addition, high concentrations of estrogens have been demonstrated in the testis, rete testis and seminal fluid in several species, suggesting that organs including the testis, efferent ductules and epididymis could be exposed to high levels of estrogens from the developing ejaculate.

Estrogen receptors are extensively distributed in adult male reproductive tracts, suggesting a possible role for endogenous estrogens. Additionally, the fetal male reproductive tract also expresses high concentrations of estrogen receptors (reviewed in [53]. This early estrogen receptor expression in developing males underlies the pronounced sensitivity to exogenous estrogen, but also indicates that fetal tissues in the male may be responsive to endogenous estrogens during development and that estrogen could play a role in normal development.

### 10.4. Use of the Estrogen Receptor 1 Knockout Mouse to Establish Roles for Estrogen in Reproductive Tract Development and Function and Fertility in Male Mice

A significant impediment to a definitive determination of whether or not there was a critical role for endogenous estrogen in the male was that estrogen was produced in various sites in males. Endogenous sources of estrogen in the male include adipose tissue, Leydig and Sertoli cells, and the germ cells of the seminiferous epithelium. Thus, it is problematical to eliminate estrogen without concomitant effects on the reproductive tract that would complicate such an experiment.

The development of the estrogen receptor 1 knockout (*Esr1*KO) mouse by Lubahn, Korach and Smithies provided a powerful and novel tool that allowed the consequences of ESR1 loss in males to be determined [56]. One striking and somewhat unexpected effect was that these animals were infertile due to impaired resorptive capacity of the efferent ductules [26]. The efferent ductules consist of a series of tubules that connect the testis to the epididymis. This small organ normally resorbs the majority of the seminiferous fluid as it passes through these structures. In the absence of normal E2/ESR1 signaling, this absorptive capacity was impaired, leading to fluid buildup in the testis, backpressure and eventually the loss of the germ cells in the seminiferous epithelium and a Sertoli cell-only phenotype in the seminiferous epithelium. Using these *Esr1*KO males, as well as other model systems, such as the aromatase knockout mouse that does not produce estrogen, additional actions of estrogen in the epididymis, as well as non-reproductive organs such as adipose tissue and the thymus, have been demonstrated [50,57,58,59].

### 10.5. Normal Estrogen Actions in the Male Reproductive Tract Require Membrane Estrogen Receptor 1

The vast majority of estrogen effects in both males and females are mediated by the main estrogen receptor, ESR1, which is located predominately in the nucleus of target tissues. These nuclear estrogen receptors (nESR1) signal through the canonical pathway involving ligand binding to their cognate receptors, dimerization of liganded receptors and translocation of ligand-receptor complexes to DNA. These complexes bind to estrogen-responsive elements in DNA, altering gene transcription and inducing the ultimate hormonal effects on the cell. In addition to nESR1, a small fraction of ESR1 protein produced (approximately 5–10%) is palmitoylated and routed to the cell membrane. This membrane ESR1 (mESR1) signals through tyrosine kinase and other pathways totally distinct from nESR1 (reviewed in [60]). For many years, the normal physiological roles of mESR1 signaling in females and males were conjectural, due in part to the lack of an effective model system that would allow signaling of the mESR1 (but not nESR1) to be blocked and the resultant phenotypic effects to be observed.

In 2014, two groups developed mouse models that maintained normal nESR1 signaling but had minimal or absent mESR1 signaling [27,28]. The nuclear-only estrogen receptor (NOER) female mice were infertile and have extensive reproductive and endocrine abnormalities, leading to the somewhat unexpected conclusion that loss of mESR1 alone could produce major reproductive abnormalities. The critical role of estrogen in the efferent ductules and other organs of the males suggested that loss of only mESR1 could potentially impact male reproduction as well. The NOER males had dramatic reproductive changes in the rete testis and efferent ductules that histologically resembled changes seen in these structures in those in *Esr1*KO mice [61]. The phenotype of NOER males was slightly milder than the *Esr1*KO, as one might expect; in contrast to the infertility of *Esr1*KO mice, NOER mice sometimes sired one litter as they reached puberty but then became infertile. These results indicate that mESR1 plays critical roles in the male reproductive tract, and that loss of membrane ESR1 signaling alone results in infertility.

## 11. 17α-E2, a Stereoisomer of 17β-E2, Increases Longevity in Male Mice

Recent years have seen rapid progress in our understanding of the role of endogenous E2/ESR1 signaling in males and resulted in identification of critical estrogen actions in normal development and function of reproductive and non-reproductive organs in males. Relatively new work on the actions of 17α-E2, a stereoisomer of the widely studied 17β-E2, suggests that this estrogen, which previously received scant attention from the research community, can have important and unprecedented effects on male longevity. This provocative and unexpected line of work clearly emphasizes the pleiotropic effects of estrogens and demonstrates that our understanding of estrogens and their effects in the male is still in its infancy.

In 2014, Harrison et al. [62] reported that feeding mice 17α-E2 incorporated into their food starting at weaning increased male median lifespan, but not that of the females. Strong and colleagues [63] reported shortly thereafter that a dose of 14.4 ppm of 17α-E2 incorporated into the chow of mice resulted in a striking 19% increase in median male lifespan. However, again 17α-E2 did not increase longevity in the females. Perhaps most significantly, a recent study [64] indicated that when 17α-E2 treatment was started at 16 months of age (equivalent to a 60-year-old human), this treatment still produced median longevity increases of 19%. Even when mice were given this treatment beginning at 20 months of age, significant increases in their median lifespan (11%) still occurred [64].

Despite intensive research, the reason for the sexually dimorphic nature of the 17α-E2 effect, where longevity increases only in males, remains unclear but is believed to result from the high circulating 17β-E2 concentrations in females, which may mask the 17α-E2 effects that can be detected after treatment of males. Another obvious question posed by the longevity increases in males treated with 17α-E2 was the mechanism of these effects. 17α-E2 has not been studied intensively, but data from several laboratories indicate that 17α-E2 has much less affinity for nuclear ESR1 than does 17β-E2 [65]. In addition, when 17α-E2 was administered to pre-pubertal or ovariectomized rodents to test its ability to produce the classic estrogenic response, 17α-E2 produced minimal increases in uterine wet weight that were far less than seen with 17β-E2 [66]. Thus, 17α-E2 has minimal effects on classic estrogenic endpoints such as increased uterine wet weight mediated through ESR1.

If 17α-E2 has minimal effects through ESR1, maybe it induces its longevity effects through another receptor. It was suggested in the early 2000s that 17α-E2 could act through another putative estrogen receptor, designated ER-X, to produce its effects [67,68]. However, this receptor was never identified, and the human genome data produced no information supporting the existence of this type of estrogen receptor. Other data indicate that 17α-E2 does not bind to the membrane G-protein estrogen receptor (GPER), so signaling through this receptor is unlikely to explain the actions of 17α-E2. Thus, despite evidence that this hormone has minimal estrogenic effects, a strong and rapidly increasing literature indicates that it produced powerful and unprecedented effects on longevity, and recent results have indicated that these effects may be mediated through ESR1. The characteristic metabolic changes documented in male mice in response to 17α-E2 were not seen in *Esr1*KO mice [69]. These data strongly suggested that 17α-E2 effects through ESR1 were at least necessary for the remarkable effects on longevity of 17α-E2. The paradox of a hormone that has historically been regarded as minimally active as an estrogen but has now been shown to produce unprecedented increases in male longevity is a fascinating one. This area will likely continue to generate more interest in the years to come as researchers seek to determine the basis of its ability to promote longevity.

Work to date has not noted any obvious effects on reproductive tract structure or function in the males treated with 17α-E2, but this has not been examined in detail. It is also not clear if there is an endogenous role of 17α-E2 in the male or female, even though available information suggests that significant amounts of 17α-E2 may be present in men and women. Clearly, we need much more information to fully understand the role of 17α-E2 in the male, as well as the female.

Three decades ago, the question of whether or not there was an important role for endogenous estrogen in the male was still being debated, although the ideas that males made estrogens, that exogenous estrogens could produce effects in males, and that males had receptors for estrogen were well established. We now know that not only is signaling through ESR1 necessary for the normal development and function of the male reproductive tract and fertility itself, but the loss of even the membrane form of ESR1 in males is sufficient to render these animals infertile. This finding is something that would have seemed inconceivable in the years before the development of the *Esr1*KO and NOER mouse began to provide tools to definitively evaluate the roles of ESR1 and the membrane component of its signaling and provide definitive answers to questions that could not be effectively answered earlier. The recent results with 17α-E2 emphasize that there is much that we do not know about the roles of estrogen in the male, despite rapid progress in this area over the past few decades. Future work in this area will likely reveal important additional insights into what estrogens do in males, and our understanding of the role of estrogens in the male is likely to keep evolving in the coming years.

## 12. Summary

Testosterone has been known to be essential for male fertility for nearly a century. Later testosterone ablation studies demonstrated that the steroid hormone was required for the development of the testis and accessory sex organs as well as completion of spermatogenesis. However, transgenic mouse models focusing on the activity of AR, GR or ERs in specific testicular cells have allowed molecular characterization of steroid hormone actions including the identification of specific targets of steroid hormone action that are required to maintain fertility. Without knockout mouse models, the non-intuitive discovery that estrogen signaling was required for fluid resorption in the efferent ductules would not have been possible. Studies employing mouse models also have identified intercellular signals that provide for collaborative support between testis cells to optimize spermatogenesis. In addition, use of mouse models allowed identification of signaling pathways and gene clusters that are regulated by steroid hormones in the testis. Together, these findings have characterized the matrix of signals required to maintain male fertility, allowing for the development of diagnostic tools to identify the causes of male infertility and treatments to correct errors that result in infertility.

## Figures and Tables

**Figure 1 ijms-24-02748-f001:**
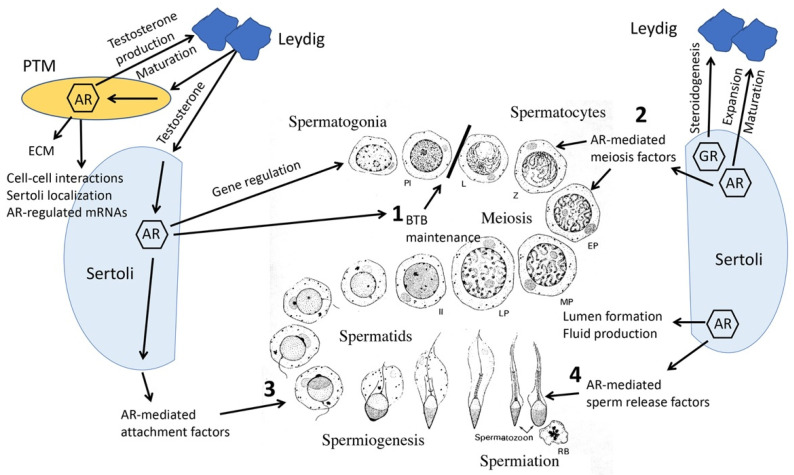
Steroid hormone-mediated signals between testicular cells maintain fertility. Androgen receptor (AR)-mediated actions in Sertoli cells support four major germ cells processes (**1**–**4**) required to maintain fertility. AR and glucocorticoid receptor (GR) in Sertoli cells support expansion and maturation of Leydig cells and steroidogenesis. Leydig cells produce testosterone that stimulates AR-mediated processes in Sertoli and peritubular myoid (PTM) cells. AR-dependent actions in PTM cells support the maturation of Leydig cells and their steroidogenesis as well as production of extracellular matrix (ECM), cell–cell interactions, Sertoli localization (positioning on the basement membrane) and AR-mediated mRNA expression in Sertoli cells. BTB, blood–testis barrier, PL, preleptotene; L, leptotene; Z, zygotene; EP, MP, LP; early, mid, late pachytene; II, meiosis II; RB, residual body.

**Table 1 ijms-24-02748-t001:** Mouse Models Defining Steroid Hormone Actions in the Male Reproductive Tract.

Mouse Model	Cell Type	Mutation/Activity	References
ARKO, T-AR^−/y^	Global	*Ar* knock out	[3,4]
iARKO	Global	Inducible *Ar* knock out	[6]
SCARKO, S-AR^−/y^	Sertoli	*Ar* knock out	[7,8]
*Ar^flox(ex1^* ^−*neo)/Y*^	Sertoli	*Ar* hypomorph	[9]
TgSCAR	Global	*Ar* gain of function	[10]
SPARKI	Global	*Gr* replacement of *Ar* exon 3	[11]
SCGRKO	Sertoli	*Gr* knock out	[12]
Ribotag	Leydig, Sertoli	Detects translated mRNAs	[13]
RiboTag-SCARKO	Sertoli	Detects translated mRNAs in *Ar* knockout	[14]
AR-C	Sertoli, Global	Classical AR activity	[15]
AR-NC	Sertoli, Global	Nonclassical AR activity	[15]
(PTM)-ARKO	Smooth Muscle (SM)	*Ar* knock out	[16]
PM-AR^−/y^	SM	*Ar* knock out	[17]
L-AR^−/Y^	Leydig	*Ar* knock out	[18]
G-AR^−/y^	Germ	*Ar* knock out	[8]
PEARKO	Prostate Epithelium	*Ar* knock out	[19]
Pes-ARKO	Prostate Epithelium	*Ar* knock out	[20]
SM-ARKO	SM	*Ar* knock out	[21]
FSP1-ARKO	Fibroblast	*Ar* knock out	[22]
dARKO	Fibroblast + SM	*Ar* knock out	[23]
ProxE-ARKO	Proximal Epididymis	*Ar* knock out	[24]
CEARKO	Principal (Epididymis)	*Ar* knock out	[25]
*Esr1*KO	Global	*Esr1* knock out	[26]
NOER	Global	Membrane *Esr1* knock out	[27,28]

Mouse models (in order of appearance in the text) having androgen receptor (AR), glucocorticoid receptor (GR), or the estrogen receptor 1 (ESR1) knocked out in all cell types (global) or specific cell types as shown. For each mouse model, the gene knock out or remaining activity is provided.

## Data Availability

Not applicable.

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
