# Peer review of "Functions of Steroid Hormones in the Male Reproductive Tract as Revealed by Mouse Models"

_ijms, 2023, doi:10.3390/ijms24032748_

Round 1
Reviewer 1 Report
The manuscript, “Functions of Steroid Hormones in the Male Reproductive Tract as Revealed by Transgenic Mouse Models” by William Walker and Paul Cooke is a review of studies that have been done to define functions of steroid hormones in the male reproductive tract. The authors focus on describing results and conclusions from studies using genetically modified mouse models of steroid hormone receptors that affect male fertility or reproductive organs. Overall, the review is well written and well-structured and summarizes the great number of studies done in this research field and will most likely be useful tool for scientists working with steroid hormone signaling and male reproduction. Some minor revisions should be done.
Specific comments
One thing that would make the manuscript clearer is to avoid the term “transgenic” when referring to knockout or knockin mouse models, since these are different techniques with different advantages and disadvantages.
Table 1 needs adjustments since the formatting is lost in the manuscript. Further the table should have a title and preferable the description above the table.
Minor comments
Line 61: “AR-mediated translated mRNAs.” Change to: Detects translated mRNAs in Ar knockout.
Figure 1: There is an “s “missing in Steroidogenesis
Line 180: Ref 12 does not seem relevant to the SCGRKO mouse model.
Line 288: The Piggyback strategy is not clearly explained or which study it refers to.
Line 408 & 421: The TFM-ARKO mouse model is not described, or should it be tfm?
Line 464: “The size of ARKO Leydig cells did not increase [32, 33].” Should it be cell population?
Line 501: GAR-/y should be floxed AR or GAR-/y (also in table 1).
Line 647: Change “chapter” to article.
Line 654: Ref 44 should most likely be removed from this place.
Line 737 knock-in mice, not transgenic.

Author Response
Reviewer 1:
We appreciate the helpful suggestions provided by the reviewer.
The manuscript, “Functions of Steroid Hormones in the Male Reproductive Tract as Revealed by Transgenic Mouse Models” by William Walker and Paul Cooke is a review of studies that have been done to define functions of steroid hormones in the male reproductive tract. The authors focus on describing results and conclusions from studies using genetically modified mouse models of steroid hormone receptors that affect male fertility or reproductive organs. Overall, the review is well written and well-structured and summarizes the great number of studies done in this research field and will most likely be useful tool for scientists working with steroid hormone signaling and male reproduction. Some minor revisions should be done.
Specific comments
One thing that would make the manuscript clearer is to avoid the term “transgenic” when referring to knockout or knockin mouse models, since these are different techniques with different advantages and disadvantages.
Response: The word “transgenic” was removed from all inappropriate places including the title.
Table 1 needs adjustments since the formatting is lost in the manuscript. Further the table should have a title and preferable the description above the table.
Response: Table 1 formatting was repaired, the list of mouse models was reorganized to match their appearance in the reorganized text and citations were updated
Minor comments
Line 61: “AR-mediated translated mRNAs.” Change to: Detects translated mRNAs in Ar knockout.
Response: Completed
Figure 1: There is an “s “missing in Steroidogenesis
Response: The figure formatting was altered when inserted into the text causing letters to be dropped from words. The figure was repaired and efforts were made to defend against problems caused by publication related format changes.
Line 180: Ref 12 does not seem relevant to the SCGRKO mouse model.
Response: Agreed, A more appropriate reference was used, now reference 12.
Line 288: The Piggyback strategy is not clearly explained or which study it refers to.
Response: The text was improved including removing the word piggyback.
Line 408 & 421: The TFM-ARKO mouse model is not described, or should it be tfm?
Response: Thank you for finding this error. TFM-ARKO was corrected to be PTM-ARKO.
Line 464: “The size of ARKO Leydig cells did not increase [32, 33].” Should it be cell population?
Response: We altered the sentence to be more clear: “The size (surface area) of ARKO Leydig cells did not increase [43, 44].”
Line 501: GAR-/y should be floxed AR or GAR-/y (also in table 1).
Response: GAR-/y was changed to G-AR-/y in the text and Table 1.
Line 647: Change “chapter” to article.
Response: Completed
Line 654: Ref 44 should most likely be removed from this place.
Response: Ref 44 was removed.
Line 737 knock-in mice, not transgenic.
Response: Completed.
Reviewer 2 Report
Walker and Cooke thoroughly reviewed the previous and recent works employing transgenic approaches to interrogate the detailed functions of steroid hormones in the male reproductive tract. This article will be a very useful reference for the field and attract broad attention from researchers interested in androgen and estrogen-mediated signaling pathways. I enjoyed reading this review and support publication in IJMS. The manuscript could be further improved by addressing the minor issues below:
Major points:
1. This review article is overall very well written but lacks a clear outline in the sections on androgen/AR. Connections or transitions are needed between subsections.
For example, from Line 29 to 399, ARKO, SCARKO, TgSCAR, SPARKI, SCGRKO, iARKO, hypomorph SCARKO, SCARKO coupled with RNA-seq, RiboTag-SCARKO, SCARKO coupled with scRNA-seq, and AR-C/AR-NC are introduced. These sections are individually clear and easy-to-follow, but it feels that the writing jumps randomly from one topic to another. The various transgenic mouse lines should be ideally elaborated in an order based on the targeted cell types or their specific functions in investigating certain AR signaling pathways.
Consistent headings and subheadings will also be helpful for the readers to follow the logic flow. For example, whilst most of the other headings describe the names of transgenic mouse lines, Line 278 indicates the importance of SCAR in meiosis. Please reorganize the subsections and revise the headings to make the flow clear to the audience.
2. Lines 36 - 47: Quote the original research articles here. More importantly, please make sure that original articles are adequately cited throughout the text.
Minor points:
1. Line 24: A more detailed introduction about androgen receptors is necessary.
2. Line 29: Change “AR receptors” to “ARs”
3. Lines 52 - 76 (Table 1): It would be helpful if the authors could show the promoters used for driving transgenic expression.
4. Lines 262 - 263: Please clarify whether these DEGs are Sertoli cell-specific or Sertoli cell-expressing genes.
5. Lines 329 - 335: References.
6. Line 803: This paragraph could be merged with the previous section on 17α-E2.
Author Response
We appreciate the helpful suggestions provided by the reviewer.
Reviewer 2:
Walker and Cooke thoroughly reviewed the previous and recent works employing transgenic approaches to interrogate the detailed functions of steroid hormones in the male reproductive tract. This article will be a very useful reference for the field and attract broad attention from researchers interested in androgen and estrogen-mediated signaling pathways. I enjoyed reading this review and support publication in IJMS. The manuscript could be further improved by addressing the minor issues below:
Major points:
- This review article is overall very well written but lacks a clear outline in the sections on androgen/AR. Connections or transitions are needed between subsections.
For example, from Line 29 to 399, ARKO, SCARKO, TgSCAR, SPARKI, SCGRKO, iARKO, hypomorph SCARKO, SCARKO coupled with RNA-seq, RiboTag-SCARKO, SCARKO coupled with scRNA-seq, and AR-C/AR-NC are introduced. These sections are individually clear and easy-to-follow, but it feels that the writing jumps randomly from one topic to another. The various transgenic mouse lines should be ideally elaborated in an order based on the targeted cell types or their specific functions in investigating certain AR signaling pathways.
Consistent headings and subheadings will also be helpful for the readers to follow the logic flow. For example, whilst most of the other headings describe the names of transgenic mouse lines, Line 278 indicates the importance of SCAR in meiosis. Please reorganize the subsections and revise the headings to make the flow clear to the audience.
Response: The text was improved with the sections being reordered to better place like-minded mouse models in the same section. New more informative and inclusive subheadings were added including sections covering Global elimination of AR, Sertoli cell-specific mouse models, AR regulated differentially expressed genes (DEGs) identified by mouse models, Peritubular (PTM), Leydig and germ cell-specific AR knockout models and knockout of AR in accessory sex organs.
- Lines 36 - 47: Quote the original research articles here. More importantly, please make sure that original articles are adequately cited throughout the text.
Response: The focus of this article is to review AR mouse models. The important foundation provided by studies of tfm rodents is not a central theme of the manuscript, thus we believe that a review providing citations of previous advances using tfm mice is appropriate.
Minor points:
- Line 24: A more detailed introduction about androgen receptors is necessary.
Response: Additional details about AR were added.
- Line 29: Change “AR receptors” to “ARs”
Response: completed
- Lines 52 - 76 (Table 1): It would be helpful if the authors could show the promoters used for driving transgenic expression.
Response: The promoters used for driving Cre and Ar genes are provided in the text for all knockout models. There is not sufficient space in the table to include all the promoters used and the citations required.
- Lines 262 - 263: Please clarify whether these DEGs are Sertoli cell-specific or Sertoli cell-expressing genes.
Response: The DEGs are now categorized as being expressed in Sertoli cells
- Lines 329 - 335: References.
Response: A reference [5] was added for the passage.
- Line 803: This paragraph could be merged with the previous section on 17α-E2.
Response: Completed.